# A method for rapid machine learning development for data mining with doctor-in-the-loop

**Neva J. Bull** [1,2] *, **Bridget Honan** [3], **Neil J. Spratt** [2,4,5], **Simon Quilty** [3,6]

1 School of Psychological Sciences, University of Newcastle, Callaghan, NSW, Australia, 2 Hunter Medical Research Institute, John Hunter Hospital, New Lambton Heights, NSW, Australia, 3 Alice Springs Hospital, Alice Springs, NT, Australia, 4 School of Biomedical Sciences and Pharmacy, University of Newcastle, Callaghan, NSW, Australia, 5 Department of Neurology, John Hunter Hospital, New Lambton Heights, NSW, Australia, 6 National Centre of Epidemiology and Population Health, Australian National University, Canberra, ACT, Australia

☯ These authors contributed equally to this work.
* neva.bull@newcastle.edu.au

**Data Availability Statement:** The study uses two human health-related data sets, both of which require Human Research Ethics Committee (HREC) approval for access and analysis, and both of which are third-party and require direct application

## Abstract

Classifying free-text from historical databases into research-compatible formats is a barrier for clinicians undertaking audit and research projects. The aim of this study was to (a) develop interactive active machine-learning model training methodology using readily available software that was (b) easily adaptable to a wide range of natural language databases and allowed customised researcher-defined categories, and then (c) evaluate the accuracy and speed of this model for classifying free text from two unique and unrelated clinical notes into coded data. A user interface for medical experts to train and evaluate the algorithm was created. Data requiring coding in the form of two independent databases of free-text clinical notes, each of unique natural language structure. Medical experts defined categories relevant to research projects and performed 'label-train-evaluate' loops on the training data set. A separate dataset was used for validation, with the medical experts blinded to the label given by the algorithm. The first dataset was 32,034 death certificate records from Northern Territory Births Deaths and Marriages, which were coded into 3 categories: haemorrhagic stroke, ischaemic stroke or no stroke. The second dataset was 12,039 recorded episodes of aeromedical retrieval from two prehospital and retrieval services in Northern Territory, Australia, which were coded into 5 categories: medical, surgical, trauma, obstetric or psychiatric. For the first dataset, macro-accuracy of the algorithm was 94.7%. For the second dataset, macro-accuracy was 92.4%. The time taken to develop and train the algorithm was 124 minutes for the death certificate coding, and 144 minutes for the aeromedical retrieval coding. This machine-learning training method was able to classify free-text clinical notes quickly and accurately from two different health datasets into categories of relevance to clinicians undertaking health service research.

to data custodians. In order to access these data, approval is first required through the NT Department of Health/Menzies School of Health Research HREC (email: ethics@menzies.edu.au), referencing this study (reference 2021-4056). Once granted HREC approval, data custodians are as follows: first, mortality data for the Northern Territory from 1980-2019 available through Australian Births Deaths and Marriages (email: agd. registrargeneral@nt.gov.au). The second set, all aeromedical retrievals in the NT from 2018-2019, is available through a combination of Careflight NT clinical retrieval database (email: helppoint@careflight.org (Top End)) and the Alice Springs Hospital Medical Retrieval and Consultation Centre (MRaCC) clinical retrieval database (email: richard.johnson@nt.gov.au).

**Funding:** The authors received no specific funding for this work.

**Competing interests:** The authors have declared that no competing interests exist.

## Introduction

Electronic health records represent a wealth of information for health service researchers, but the utility is limited by the challenges of extracting meaningful information from vast datasets of unstructured text [1]. Machine learning (ML) tools can be used to classify free text from clinical notes into categories, with useful clinical and research applications.

ML has been used to classify cases into a medical speciality type based on clinical notes from inpatient and outpatient healthcare encounters [2]. Similarly, ML has been used to categorise healthcare encounters as either related or not related to falls, to inform clinical and health service interventions [3]. In these examples, ML tools have utilised natural language processing techniques that require significant computer programming and data science expertise, which may be beyond the scope of clinician researchers. Furthermore, the ML tool must first learn from a labelled training data set, which may be difficult, time-consuming, or expensive to obtain [4–6]. There have been recent software developments that make ML readily available to a broader range of low-resource research challenges [7], although such readily available tools have yet to appear as mainstream research tools.

The Human-in-the-Loop (HITL) approach incorporates human skills and expertise into ML processes, including creating labelled datasets [8, 9]. Compared to an automated approach, this interactive approach may achieve greater accuracy with fewer training labels because of the human expert capacity to identify patterns from relatively few samples [10].

Investigators requiring bespoke classification tasks of large-scale datasets must either create their own training dataset or use previously labelled datasets. For a research project investigating the impact of heat waves on specific health outcomes, there were no published examples of use of ML tools to classify prehospital aeromedical triage notes into medical specialty type. Furthermore, previous attempts to classify death certificate records into cause-of-death categories have achieved limited accuracy due to high number of categories and automated approaches without clinician input.

The objective of this study was to assess the accuracy and speed of the HITL ML methodological approach for clinician researchers to classify clinical notes into customised categories for future health services research purposes.

## Methods

### Study design

This is a validation study of a ML algorithm to extract clinical categorical data from unstructured text. Two distinct data sources and categorisation tasks from residents of the Northern Territory (NT) of Australia were used. The success of the method was measured in terms of overall accuracy, sensitivity, specificity, and time taken to achieve accuracy over 90% in both datasets.

### Data sources

Access to the data was approved by the Human Research Ethics Committee of the Northern Territory Department of Health and Menzies School of Health Research. All data had been fully anonymized before access and the ethics committee waived the requirement for informed consent.

Dataset 1 was a 40-year mortality database from 1980 to 2019 from the Northern Territory Births, deaths and marriages registry. The variables included age, sex, Indigenous status (self-reported), location at death, usual residential address, and cause of death as verbatim hand-recorded on the death certificate and then transcribed electronically into the database.

**Table 1. Examples of natural language in (a) mortality dataset, (b) aeromedical retrieval dataset.**

| Cause of Death text | Reason for retrieval text |
|---|---|
| Part 1 (a) days Part 1 (b) years (Part 1 (a) circulatory shock; cardiovascular failure Part 1 (b) hepatic malignancy) | MVA #femur |
| Parkinson's Disease (5 Years);Respiratory Arrest (4 Minutes);Vomiting (10 Months);Aortic Stenosis (10 Years) | Bite to toe |
| Metastatic colorectal cancer (3 years) | Bilat flank pain, oliguria, acidosis? cause |
| Septaecemia;Left above knee amputation, infected stump, end-stage renal failure, type 2 diabetes. | AKI |
| INQUEST DISPENSED WITH [XXX DATE–CORONER'S NAME]** - Died from Multiple Injuries received in a motorcycle accident where deceased was the rider.** | HD pt for I and D |
| Spontaneous intracerebral haemorrhage (Hours);Atrial fibrillation on warfarin (Years);Hypertension (Years);Type 2 diabetes (Years) | Detiorating mental state, Schitz |
| INQUEST DISPENSED WITH [XXX DATE–CORONER'S NAME]** Died from natural causes being Coronary Atherosclerosis. | PPROM 29+5/40 |

Spelling and structural errors are a reflection of dataset

** episodes de-identified in their presentation due to medicolegal sensitivities

The aeromedical retrieval dataset included every aeromedical retrieval in the NT between January 2018 –August 2020. Aeromedical retrieval data is generated when a remote community healthcare worker refers a patient requiring transport to a higher level of hospital care. In this region, transport is conducted by fixed wing aircraft and prioritisation and clinical decision-making about transport is undertaken by medical specialists in the medical retrieval coordination centre.

The outcome of interest was stroke category (ischaemic stroke, haemorrhagic stroke or no stroke). This text was era-dependent with medical language and diagnostics changing over the 40-year period. There were two language structures–clinical (when recorded by the certifying doctor) and legal (when reported by the coroner). It was unstructured natural language and had numerous spelling and typographical errors. Examples of the cause of death text are shown in Table 1.

The variables included were date of retrieval tasking, age, triage priority (1–6 in order of urgency, with 1 representing highest level emergency and 6 representing non-urgent), whether a doctor was tasked to accompany the retrieval, and clinical reason for retrieval recorded in shorthand by the doctor receiving the initial emergency phone call triggering the retrieval. Examples of the reason for retrieval are shown in Table 1.

## Classification outcomes

The coding task chosen for cause-of-death text was a classification into 3 mutually exclusive categories: Ischaemic, Haemorrhagic or Not (no mention of stroke). This was not restricted to the word "Stroke" and included anything that could be reasonably interpreted by a clinical expert as meaning stroke, such as CVA, cerebrovascular accident, middle cerebral artery infarct or intracerebral haemorrhage.

The coding task for the aeromedical retrieval was coding reason-for-retrieval text into a 5-category classification of in-hospital receiving specialty destination for each retrieval. The original research objective required coding of this data into broad clinical categorisation of hospital speciality destinations, as the pre-hospital 'reason for retrieval' was recorded pre-diagnostically. The specialty destination categories were Medical, Surgical, Trauma, Obstetric and Psychiatric (where trauma is a sub-specialty of surgery but a discrete entity in terms of aetiology of injuries sustained and subsequent retrieval services requested).

## Creation of validation datasets

Two medical specialists developed a set of rules to define coding for each dataset. For instance, where there was mention of both ischaemic and haemorrhagic stroke in any given case, 'Ischaemic' was labelled, unless there was a three-month period between events where the haemorrhagic stroke was clearly recorded as the final event. Subdural and subarachnoid haemorrhages were labelled as 'not', as was intracranial haemorrhage, not further specified (as these events were often associated with a fall or trauma), however intracerebral haemorrhage (which was presumed to be an intra-parenchymal event) was labelled as 'Haemorrhagic'. Septic emboli to the brain were labelled 'Not'.

Each medical specialist independently labelled 300 cases for each validation dataset. If there was disagreement between the specialists, they attempted to resolve by consensus or deferred to a third independent medical specialist for a final decision.

## Preprocessing and selection of ML algorithms

The ML implementation was written in C# using.NET Core. The database used was SQL Server and the web-application was written in PHP and Javascript. Data was analysed using the R programming language. An open-source machine learning framework with nine potential candidate algorithms for multiclass classification of text data was used (ML.NET). The preferred algorithm for each task was selected during a process of semi-automated experimentation with train/test sets labelled by a single clinical expert.

To minimize over-fitting, comparisons of the accuracy of all algorithms were repeated with systematic exclusions of one or more of the available supplementary features (such as age or gender) until a minimal feature set was found that did not compromise accuracy. Cases with missing supplementary features were included.

Parameter experimentation was also performed for each algorithm. Parameters that were tested were: inclusion of stop words, punctuation, numbers and varying the length of n-grams and char-grams. Each experiment was performed with an 80–20 train-test split using a consistent seed. The best algorithm for each task was chosen based on micro- and macro-accuracy.

## Label-train-evaluate loop

A custom user interface (UI) in the form of a web-application was developed that facilitated the investigators to complete the workflow remotely.

The method maintained 3 separate splits of the whole dataset, these were a test set that did not change throughout the study, was not included in training and was not visible to the labellers, a training set and a prediction set. The prediction set comprised the remainder of the dataset after exclusion of the training and test sets. Training only occurred on the training set. Predictions were made against both the test set (to calculate interim accuracy metrics) and the remaining dataset, after exclusion of the training set.

Balanced test sets were developed by initially labelling cases at random and then using text search on keywords to add additional cases in the rare categories. The test sets were created by two medical experts who labelled each case independently, and disagreements were resolved by consensus. If consensus could not be reached, a third independent medical expert made the final decision.

The method provided a mechanism to select predictions for expert labelling. These newly labelled data were then moved to the training set and excluded from further prediction.

Each of the datasets were trained and evaluated by two medical experts in a "label–train–evaluate" loop (Fig 1).

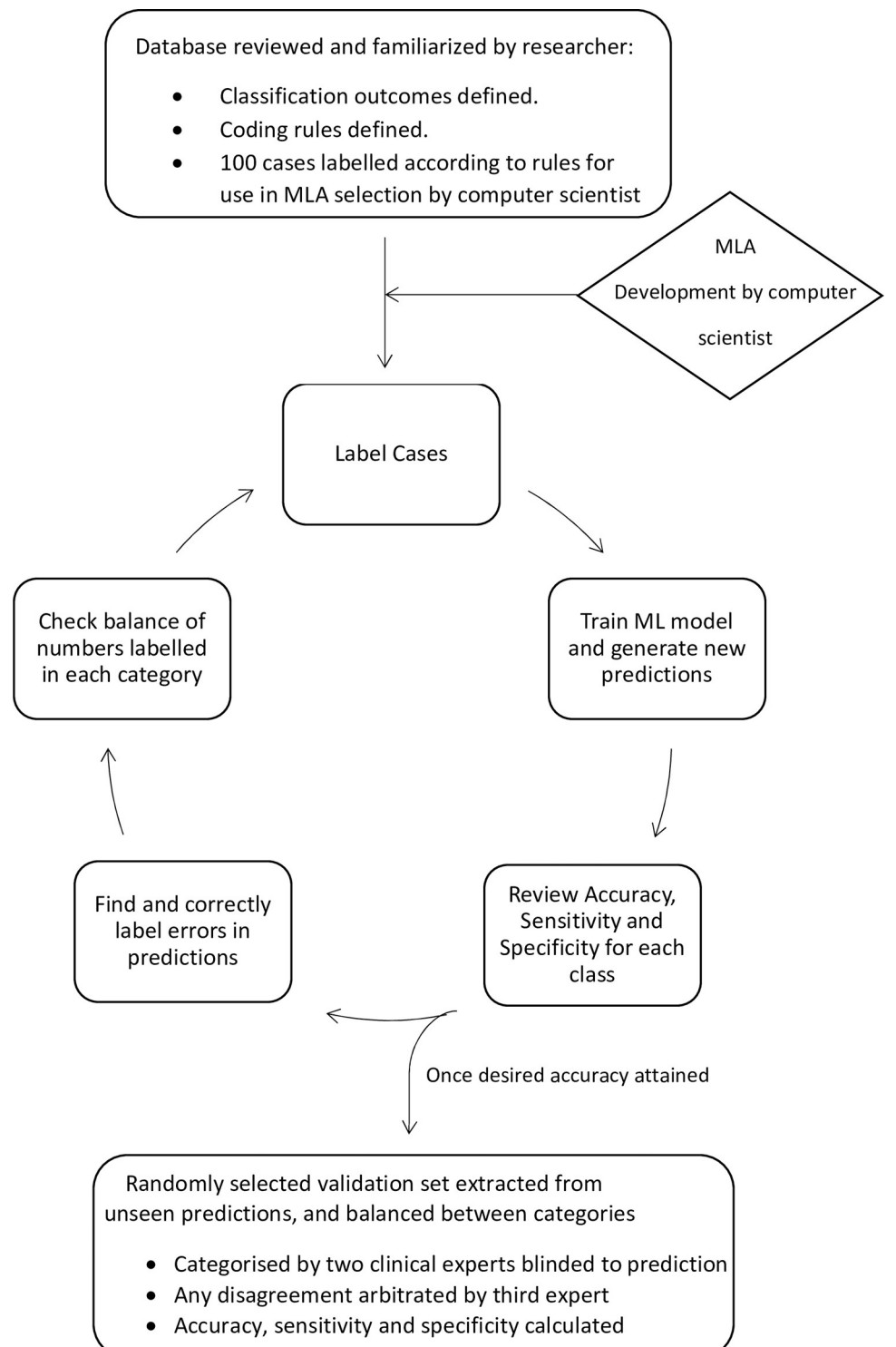

**Fig 1. Label-train-evaluate loop.**

After each round of labelling training data, the medical experts triggered retraining of the model on the server by pressing a button on the UI. The UI then displayed new predictions and accuracy metrics, in about 12 seconds.

The UI facilitated predictions to be evaluated either at random, by text search or by sorting on confidence scores and could be filtered based on predicted category. The medical experts could choose to ignore correct predictions with high confidence scores, confirm correct prediction with low confidence scores or label incorrect predictions. Confirmed and relabelled predictions were moved to the training set and were not included in future predictions.

When consistent misclassifications were detected, keywords were used to find and label additional similar cases to be added to the training set. Predictions could also be sorted on their confidence score. This workflow was designed with the aim of producing rapid and maximal improvement of the model by focusing labelling efforts on major misclassifications. When high confidence errors were minimal, cases in each category were listed in ascending order of confidence allowing for more difficult cases to be added to the training set.

## Validation set

After the 'label-train-evaluate' loop was concluded, a validation set was selected for labelling by two medical experts who were blinded to the category predicted by the ML model. The validation set was balanced between groups by randomly selecting equal numbers of cases based on the category inferred by the ML model.

## Results

### Optimised models

After experimentation, the decision tree algorithm (light gradient boosted machine) was selected for the stroke classification task and a linear algorithm (stochastic dual coordinated ascent maximum entropy) was selected for the aeromedical retrieval classification task.

For the death certificate data, stop words, punctuation and numbers were removed. For the aeromedical retrieval data, punctuation, stop words and numbers were included in the featurisation. In both cases n-grams were restricted to a maximum of 2 and char-grams a maximum of 3. Categorical variables were one hot encoded and continuous variables were normalised. Cases with missing free text were excluded.

There were 34,034 cases in the cause of death dataset. The text had a mean length of 134 characters (range 8–591). Labelling and training took the clinicians 124 minutes to complete. The macro accuracy of the ML method in classifying each case as "Ischaemic", "haemorrhagic" or "Not stroke" was 94.7% in the validation set (Table 2).

**Table 2. Dataset 1 accuracy.**

|  |  |  | Truth (Validation set) |  |  |  |
|---|---|---|---|---|---|---|
|  |  | Isch | Haem | Not | TOTAL | Sens |
| **Predicted** | Isch | 49 | 0 | 1 | 50 | 0.98 |
|  | Haem | 2 | 43 | 5 | 50 | 0.86 |
|  | Not | 0 | 0 | 50 | 50 | 1 |
|  | TOTAL | 51 | 43 | 56 |  |  |
|  | Spec | 0.98 | 1 | 0.94 | **Accuracy = 0.947** |  |

The trajectory of sensitivity and specificity of the ML (calculated against the test set) as the human experts coded more data is shown in Fig 2.

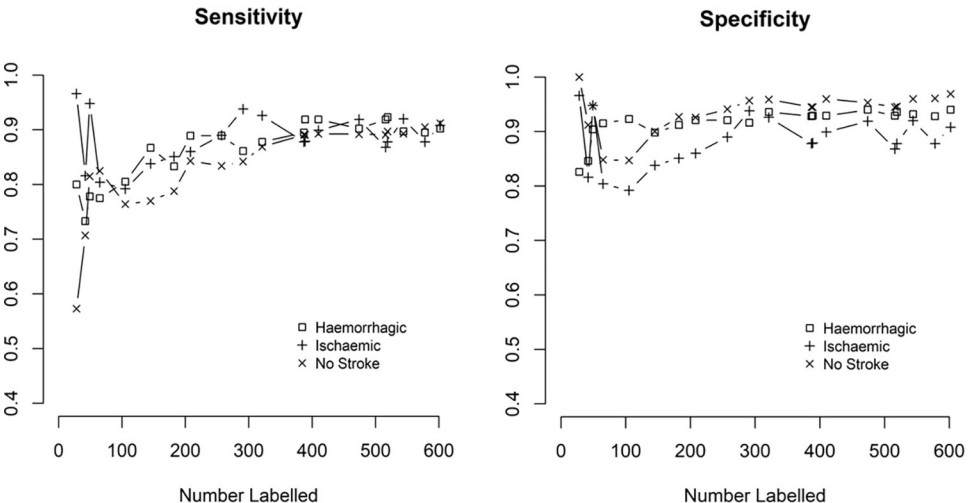

**Fig 2. Iterative sensitivity (left) and specificity (right) for 'mention' of stroke in death certificate data.**

There were 12,039 cases in the aeromedical retrieval dataset. The reason-for-retrieval comprised short text strings with an average length of 20 characters (range: 1–94). The human experts labelled a total of 550 cases. Time to complete was 144 minutes. The macro accuracy of the ML categorisation was 92.4% (Table 3.)

As demonstrated in Fig 4, the overall accuracy of ML on both datasets improved iteratively at approximately the same rate.

For distinct diagnostic categories (Haemorrhagic Stroke, Psychiatric and Obstetric) there was high intra-class accuracy even with small numbers labelled (Fig 3). Fig 3 also demonstrates that distinguishing between medical, surgical and trauma, which were subtly different, had lower sensitivity and specificity and took more labelling to increase these metrics.

Despite the distinctive linguistic structures of each of the two data sets, the method allowed for similarly rapid improvement in accuracy with training iterations as demonstrated in Fig 4.

## Discussion

In this study, historically collected free-text data from healthcare records was classified into customised categories for use in health services research projects using a human-in-the-loop interactive ML methodology. The HITL labelling process took 124 minutes and was 94.7%

**Table 3. Dataset 2 accuracy.**

| | | Truth (validation set) | | | | | | |
|---|---|---|---|---|---|---|---|---|
| | | Medical | Psychiatric | Obstetric | Trauma | Surgical | TOTAL | Spec. |
| | Medical | 46 | 0 | 0 | 2 | 2 | 50 | 0.98 |
| **Predicted** | Psychiatric | 0 | 50 | 0 | 0 | 0 | 50 | 1.00 |
| | Obstetric | 2 | 0 | 48 | 0 | 0 | 50 | 0.99 |
| | Trauma | 4 | 0 | 1 | 43 | 2 | 50 | 0.96 |
| | Surgical | 6 | 0 | 0 | 0 | 44 | 50 | 0.97 |
| | TOTAL | 58 | 50 | 49 | 45 | 48 | | |
| | Sens. | 0.92 | 1.00 | 0.96 | 0.86 | 0.88 | **Accuracy = 0.924** | |

The trajectory of sensitivity and specificity as the human experts coded additional data is shown in Fig 3.

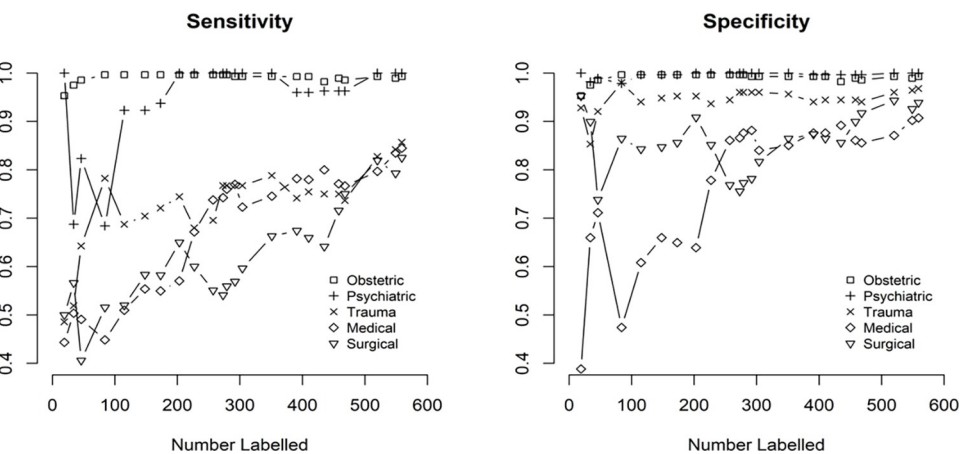

**Fig 3. Iterative sensitivity (left) and specificity (right) of aeromedical retrieval data.**

accurate in classifying death certificate data into 3 categories related to stroke diagnosis. For aeromedical retrieval triage data, the HITL labelling process took 144 minutes and was 92.4% accurate in classifying cases into 5 categories related to medical specialty type.

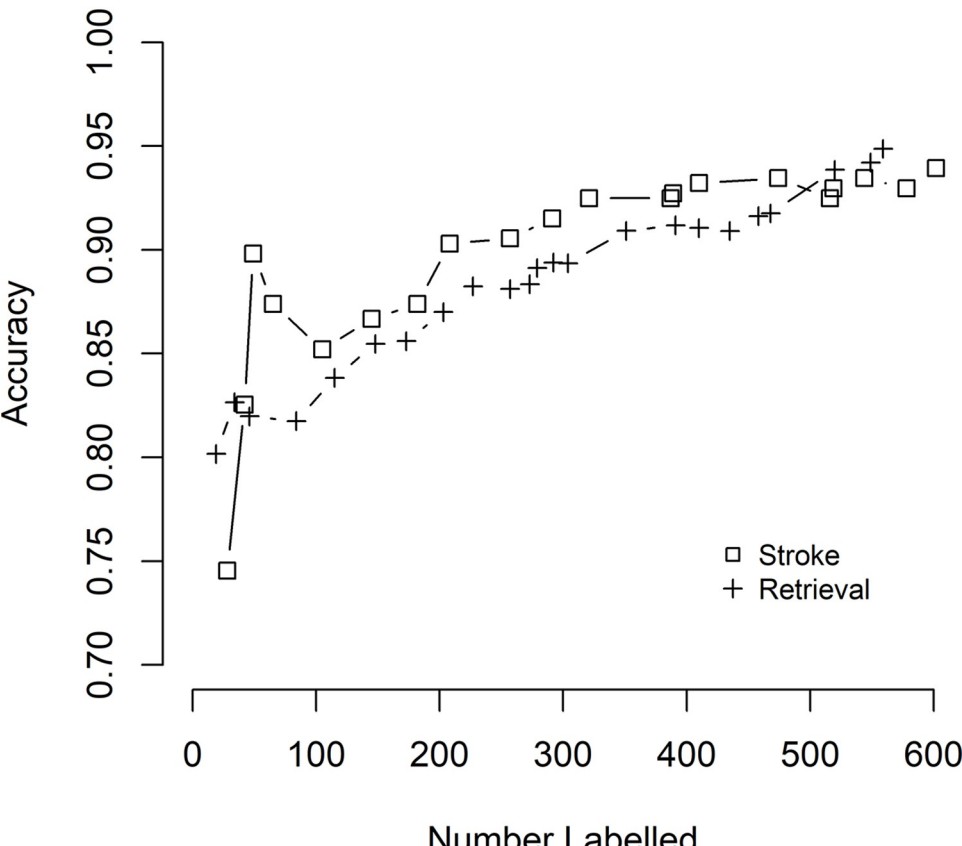

**Fig 4. Accuracy of stroke classification and retrieval classification as a function of number labelled.**

As our study demonstrates, the challenge of coding complex natural language databases can have simple solutions using well-developed frameworks for automating ML algorithms (we used ML.NET). This allowed us to use our own understanding of the datasets that we were applying our research question to, and then iteratively train the model until we were satisfied it met our accuracy requirements. Prior to the availability of off-the-shelf software, the implementation of ML to manipulate data in such a way required high-level software engineers. With the advent of AutoML and in combination with HITL, our study proves the democratising of ML that will make such techniques available to laboratories that may not have had sufficient funding to employ ML specialists.

Previous studies that classify death certificate records into International Classification of Disease (ICD-10) diagnostic codes have not yielded the accuracy of this study [11–13], and thus were not suitable for the task of identifying stroke-related cases from a jurisdictional death registry database. Challenges to accuracy in this field include the high number of categories, unbalanced class frequencies, non-conventional language, and abbreviations, and crossover between categories. The interactive HITL approach allows the clinician to define and customise a small number of categories that are relevant to clinical, health services or research needs. Furthermore, allowing the content-expert to select cases in the training set for labelling may achieve better accuracy with smaller samples, due to the efficiency of human pattern-recognition compared to automated processes.

ML techniques have previously been applied to prehospital triage data, but has utilised numerical or categorical variables, such as heart rate or temperature rather than free text in clinical notes [14]. A supervised ML tool that classified clinical notes into medical specialty types achieved a high degree of accuracy. In this study, 2 expert clinicians labelled 431 clinical notes by medical specialty category. The authors do not report on the time taken for the clinical experts to complete the labelling, nor the time for computer programmers and data scientists to compare and select the best-performing classifier technique.

Furthermore, it is unknown how this MLA would perform when applied to novel datasets. Given the variations in language and abbreviations in different health care settings, machine learning tools will perform best when applied to a dataset that is derived from the same source as the training dataset [15]. For this reason, researchers may prefer to create a custom ML tool for each new project, rather than utilising a ML algorithm that was trained on a dissimilar dataset.

## Conclusion

An interactive approach to machine learning that uses readily available off-the-shelf ML software and medical experts as the "human-in-the-loop" for training data sets was used to rapidly and accurately classify free text from healthcare records into customised categories.

## Author Contributions

**Conceptualization:** Neva J. Bull, Simon Quilty.

**Data curation:** Neva J. Bull.

**Formal analysis:** Neva J. Bull.

**Investigation:** Bridget Honan, Simon Quilty.

**Methodology:** Neva J. Bull.

**Software:** Neva J. Bull.

**Supervision:** Neil J. Spratt.

**Writing – original draft:** Neva J. Bull, Bridget Honan, Simon Quilty.

**Writing – review & editing:** Neva J. Bull, Bridget Honan, Neil J. Spratt, Simon Quilty.

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
