## [Decision Letter · Decision Letter 0]

26 Dec 2022

PONE-D-22-29006A method for rapid machine learning development for data mining with Doctor-In-The-LoopPLOS ONE

Dear Dr. Bull,

Thank you for submitting your manuscript to PLOS ONE. After careful consideration, we feel that it has merit but does not fully meet PLOS ONE’s publication criteria as it currently stands. Therefore, we invite you to submit a revised version of the manuscript that addresses the points raised during the review process. Please address all the concerns from the reviewers and the editor and revise your manuscript accordingly.

We look forward to receiving your revised manuscript.

Kind regards,

Ernesto Iadanza

Academic Editor

PLOS ONE

Journal Requirements:

Additional Editor Comments:

The authors present a decision-making tool that involves clinical experts to participate in the training of off-the-shelf machine learning algorithms aimed at classifying free-text medical notes. The article is well written and easy to follow, and the tool presented seems to offer a means for clinicians to get involved in the crafting of the tool and in improving its performance, which may eventually help with the clinical practicality. However, the proposed framework can hardly be claimed as original work as it is essentially an active learning approach applied to the problem of free-text classification. The tool could in a way be regarded as novel, but not a research novelty. The authors need to clarify and support what they propose to be truly original.

Reviewers' comments:

Reviewer's Responses to Questions

**Comments to the Author**

1. Is the manuscript technically sound, and do the data support the conclusions?

Reviewer #1: Partly

2. Has the statistical analysis been performed appropriately and rigorously? 

Reviewer #1: N/A

3. Have the authors made all data underlying the findings in their manuscript fully available?

Reviewer #1: No

4. Is the manuscript presented in an intelligible fashion and written in standard English?

Reviewer #1: Yes

5. Review Comments to the Author

Reviewer #1: While the manuscript presents a system of potential clinical application, it lacks originality. The proposed training/evaluation protocol has extensively been used and its merit in improving the performance of machine learning algorithms has been thoroughly discussed in many published studies. Additionally, the idea of involving human experts in the training phase of a machine learning based decision-making tool is not new and links to Active Learning. It is understood that such a tool may have a clinical practicality and involving clinicians in its development may contribute to its performance improvement, it remains unclear however if the originality of the work is the tool itself or the framework to develop it.

6. PLOS authors have the option to publish the peer review history of their article (what does this mean?). If published, this will include your full peer review and any attached files.

Reviewer #1: **Yes: **Kais Gadhoumi, PhD

---

## [Author Response · Author response to Decision Letter 0]

21 Mar 2023

The originality of our paper is that it demonstrates the ready use of open source AutoML frameworks (in our case ML.NET), which in essence is proof of the democratization of ML techniques. We have now highlighted this in the discussion (paragraph 2, Discussion). 

The result is that it is now possible to use ML to assist clinical research without postgraduate level ML expertise. The implications of this are significant. Principally, it brings the utility of ML in processing large datasets to laboratories that may not have sufficient funding to employ ML specialists. A fundamental understanding of statistics is still required, and we assume that clinical researchers have this.

The method we describe differs from typical active learning in that there was no automation of selection of new datapoints to label. We demonstrate that it is both sufficient and efficient to use the real intelligence of humans (in our case; Doctors) to determine which additional cases to label. 

Of note, we chose to demonstrate that efficiency was reproducible across two different data sets and categorisation tasks. We did not do a direct comparison to any of the myriad of existing active learning techniques, as it was not conceived as an attempt to advance the field of active learning from a computer science perspective. Similarly, we elected not to include in the manuscript a lengthy review of active learning and Human-In-The-Loop strategies as the intended audience is scientists who have a need for categorization of large datasets but don’t come from a computer science background. For these reasons we also chose to avoid Deep Learning. We may have achieved superior results using a transformer such as BERT, however this would add significant complexity to the software development step – defeating the purpose of creating an easily reproducible method that doesn’t require advanced ML and computer science skills. 

In essence, the manuscript is a description of how ML can be implemented quickly, cheaply, achieve high accuracy using the existing resources of a typical research laboratory. This goes to the Utility requirement of PLOS ONE’s submission guidelines, the manuscript presents experimental data demonstrating the Validation requirement. The tool was developed using open-source libraries and the manuscript provides sufficient detail for the tool to be replicated in any programming framework capable of ML. This ensures Availability of the method. As for Novelty, our reviews of the literature did not find this combination of techniques previously reported.

---

## [Decision Letter · Decision Letter 1]

13 Apr 2023

A method for rapid machine learning development for data mining with Doctor-In-The-Loop

PONE-D-22-29006R1

Dear Dr. Bull,

We’re pleased to inform you that your manuscript has been judged scientifically suitable for publication and will be formally accepted for publication once it meets all outstanding technical requirements.

Kind regards,

Ernesto Iadanza

Academic Editor

PLOS ONE

Additional Editor Comments (optional):

Reviewers' comments:

Reviewer's Responses to Questions

**Comments to the Author**

1. If the authors have adequately addressed your comments raised in a previous round of review and you feel that this manuscript is now acceptable for publication, you may indicate that here to bypass the “Comments to the Author” section, enter your conflict of interest statement in the “Confidential to Editor” section, and submit your "Accept" recommendation.

Reviewer #1: All comments have been addressed

2. Is the manuscript technically sound, and do the data support the conclusions?

Reviewer #1: Yes

3. Has the statistical analysis been performed appropriately and rigorously? 

Reviewer #1: Yes

4. Have the authors made all data underlying the findings in their manuscript fully available?

Reviewer #1: No

5. Is the manuscript presented in an intelligible fashion and written in standard English?

Reviewer #1: Yes

6. Review Comments to the Author

Reviewer #1: (No Response)

7. PLOS authors have the option to publish the peer review history of their article (what does this mean?). If published, this will include your full peer review and any attached files.

Reviewer #1: **Yes: **Kais Gadhoumi, PhD

---

## [Editor Report · Acceptance letter]

18 Apr 2023

PONE-D-22-29006R1 

A method for rapid machine learning development for data mining with Doctor-In-The-Loop. 

Dear Dr. Bull:

I'm pleased to inform you that your manuscript has been deemed suitable for publication in PLOS ONE. Congratulations! Your manuscript is now with our production department. 

Kind regards, 

on behalf of

Dr. Ernesto Iadanza 

Academic Editor

PLOS ONE